# Flecainide Specifically Targets the Monovalent Countercurrent Through the Cardiac Ryanodine Receptor, While a Dominant Opposing Ca^2+^/Ba^2+^ Current Is Present

**DOI:** 10.3390/ijms26010203

**Published:** 2024-12-29

**Authors:** Jana Gaburjakova, Michaela Domsicova, Alexandra Poturnayova, Marta Gaburjakova

**Affiliations:** Institute of Molecular Physiology and Genetics, Centre of Biosciences, Slovak Academy of Sciences, Dubravska cesta 9, 840 05 Bratislava, Slovakia; jana.gaburjakova@savba.sk (J.G.); michaela.domsicova@savba.sk (M.D.); alexandra.poturnayova@savba.sk (A.P.)

**Keywords:** flecainide, antiarrhythmic drug, ryanodine receptor, countercurrent, arrhythmia, catecholaminergic polymorphic ventricular tachycardia

## Abstract

Catecholaminergic polymorphic ventricular tachycardia (CPVT) is a highly arrhythmogenic syndrome triggered by stress, primarily linked to gain-of-function point mutations in the cardiac ryanodine receptor (RyR2). Flecainide, as an effective therapy for CPVT, is a known blocker of the surface-membrane Na^+^ channel, also affecting the intracellular RyR2 channel. The therapeutic relevance of the flecainide-RyR2 interaction remains controversial, as flecainide blocks only the RyR2 current flowing in the opposite direction to the physiological Ca^2+^ release from the sarcoplasmic reticulum (SR). However, it has been proposed that charge-compensating countercurrent from the cytosol to SR lumen plays a critical role, and its reduction may indeed suppress excessive diastolic SR Ca^2+^ release through RyR2 channels in CPVT. Monitoring single-channel properties, we examined whether flecainide can target intracellular pathways for charge-balancing currents carried by RyR2 and SR Cl^−^ channels under cell-like conditions. Particularly, the Tris^+^ countercurrent flowed through the RyR2 channel simultaneously with a dominant reverse Ca^2+^/Ba^2+^ current. We demonstrate that flecainide blocked the RyR2-mediated countercurrent without affecting channel activity. In contrast, the SR Cl^−^ channel was completely resistant to flecainide. Based on these findings, it is reasonable to propose that the primary intracellular target of flecainide in vivo is the RyR2-mediated countercurrent.

## 1. Introduction

Flecainide is a class Ic antiarrhythmic agent that is predominantly indicated in patients for the prevention of atrial fibrillation and supraventricular tachycardias (reviewed in [1]). However, due to its proarrhythmic risks under some clinical circumstances, flecainide is recommended only for carefully selected groups of patients. Because ventricular proarrhythmic effects only rarely occur when the function of the left ventricle is preserved [2], flecainide is a drug of choice for the management of patients diagnosed with a life-threatening, stress-induced arrhythmogenic disorder in structurally normal hearts known as catecholaminergic polymorphic ventricular arrhythmias (CPVT) [3,4,5,6]. CPVT is predominantly identified in individuals who are carriers of a dominant point mutation in the cardiac ryanodine receptor (RyR2) [7,8,9,10,11]. This protein functions as an intracellular Ca^2+^ channel residing in the membrane of the cardiac sarcoplasmic reticulum (SR) and plays a critical role in a massive Ca^2+^ release from the SR into the cytoplasm [12,13,14]. A transient increase in intracellular Ca^2+^ is essential for triggering cardiac muscle contraction, and the RyR2 channel must be tightly regulated to maintain proper spatiotemporal patterns of Ca^2+^ signaling. CVPT-linked mutations in the RyR2 channel result in impaired Ca^2+^ release in diastole, particularly under adrenergic stress, leading to delayed afterdepolarizations (DADs) of the plasma membrane, which might trigger CPVT when they reach the action potential threshold [9,10,15,16].

The antiarrhythmic action of flecainide is primarily related to the well-characterized blockage of the voltage-gated Na^+^ channel (Nav1.5) [17,18], thereby decreasing plasma membrane excitability. Given that the Na^+^ channel is located on the cell surface and has a high affinity for flecainide [19], this channel is inevitably the first target of flecainide as it enters the cardiac muscle cell [20]. However, in the context of CPVT, the sum of experimental evidence argues for an intricate therapeutic effect of flecainide, likely involving inhibition of the intracellular RyR2 channel as well [21,22,23]. The role of the RyR2 channel in flecainide’s action is widely debated (reviewed in [20,24,25]). Although single-channel data demonstrate a partial blockage of the RyR2 current by flecainide, resulting in a reduced RyR2 activity and occurrence of partial closings to a subconductance state, this effect has been observed only when the current through the channel flowed in the direction opposite to the physiologically relevant SR lumen-to-cytosol flow [21,22,23,26,27,28,29]. Initially, this finding negated the participation of the RyR2 channel in the antiarrhythmic action of flecainide, but this perspective has since been revisited. The main reason was that to sustain Ca^2+^ release from the SR for more than a few milliseconds, SR countercurrents are critical to balance the charge released through the RyR2 channel [30]. In addition to SR K^+^ and Cl^−^ channels as potential contributors to the charge-compensating machinery [31,32,33], the RyR2 channel has been proposed to conduct its own countercurrent, being only weakly cation-selective [30]. Such arguments prompted one possible concept that the blockage of the RyR2 countercurrent in the cytosol-to-SR lumen direction may result in the suppression of Ca^2+^ release [30], particularly during diastole when the activity of RyR2 channels harboring CPVT mutations is aberrant [28,29,34]. In connection with this point, Bannister and his colleagues developed a testable hypothesis about the contribution of the partial blockage of the RyR2-mediated countercurrent in flecainide’s therapeutic action [20]. Recently, this idea received support from Steer et al. [35], who demonstrated that the flecainide-induced decrease in the frequency of SR Ca^2+^ waves generated by Ca^2+^ release through RyR2 channels was strongly affected when SR countercurrents in permeabilized cardiomyocytes were manipulated.

Here, we use single-channel recordings of RyR2 channels from the rat heart to evaluate the power of flecainide to block the RyR2-mediated countercurrent under cell-like conditions, where Tris^+^ countercurrent coexisted with the dominant Ca^2+^/Ba^2+^ current flowing in the physiologically relevant direction. Furthermore, we tested whether SR Cl^−^ channels, as the potential source of charge-compensating currents, might also be affected by flecainide. SR K^+^ channels were not of our interest as Bannister et al. [28] have already demonstrated their resistance to flecainide’s action. Notably, flecainide is thought to reach the binding site located in the conductive pore of the voltage-gated Na^+^ channel [36], either directly from the cytosolic face or indirectly from the membrane lipid phase [37], using hydrophobic pathways termed fenestrations [36]. Since it has been proposed that fenestrations may have pharmacological significance also for other ion channels, we assessed whether flecainide (despite being 99% protonated at physiological pH [18]) might specifically interact with the lipid membrane under our single-channel experimental conditions. For this purpose, we employed the quartz crystal microbalance (QCM), which is a highly sensitive instrument suitable to monitor very small mass changes on the surface of a quartz crystal coated with a lipid membrane [38,39,40,41].

## 2. Results

### 2.1. Effect of Flecainide on Function of the RyR2 Channel When the Amplitude of Tris^+^ Countercurrent Through the Channel Was Manipulated

RyR2 channels were incorporated into BLMs and recorded under asymmetric conditions at 0 mV, with 1 mM luminal Ca^2+^ (a physiological level) mixed with 20 mM luminal Ba^2+^ serving as the main charge carriers from SR lumen to cytosol. Since the RyR2 channel is also permeable to Tris^+^ cations (G~17 pS, [42]), a Tris^+^ gradient in the cytosol-to-SR lumen direction was established to simulate cell-like conditions during SR Ca^2+^ release when a countercurrent in the opposite direction to the main Ca^2+^ current is expected to be feasible. Notably, when membrane voltage ranging from −10 mV to +20 mV was applied to determine the RyR2 conductance (G), the flow of Ca^2+^ and Ba^2+^ in the physiologically relevant SR lumen-to-cytosol direction remained consistently favored. Among the monovalent cations permeant through the RyR2 channel [42], Tris^+^ was selected instead of physiologically relevant K^+^ because crude SR microsomes, used to reconstitute RyR2 channels, also contain various SR K^+^ channels. As a result, in the presence of opposing K^+^ and Ca^2+^/Ba^2+^ gradients, we would have observed not only RyR2 channels but also K^+^ channels, which was undesirable. To collect sufficient RyR2 openings for the accurate calculation of channel G and to enable flecainide to demonstrate solely its blocking action, frequently manifested as a decrease in RyR2 activity (P_O_), 1–3 mM ATP was added to the cis compartment to achieve moderate channel activation (P_O_ ranging from 0.20 to 0.57). Notably, Salvage et al. [43] recently reported that under low-activity conditions (P_O_ < 0.08), flecainide paradoxically induced RyR2 activation. In our study, this mode of action was not allowed to occur because RyR2 channels were moderately activated by ATP, substantially favoring the blocking effect of flecainide. Since RyR2 channels incorporate into the BLMs in the fixed orientation [44], all recorded channels responded to cytosolic ATP, indicating that the RyR2 cytosolic face was oriented toward the cis compartment. Given that the site of action for flecainide is only accessible from the RyR2 cytosolic side [27], flecainide was applied exclusively to the cis compartment.

As a first step, we investigated the potential blocking effect of flecainide on the RyR2-mediated countercurrent when a 125/10 mM Tris^+^ (cis/trans) gradient was established. To obtain reliable control data for the normalization of the G values, we began by recording channel activity when equal concentrations of Tris^+^ on both sides of the BLM were present (10/10 mM). The Tris^+^ concentration was then increased to 125 mM by perfusing the cis compartment, followed by the cis application of flecainide in increasing concentrations from 2 to 200 µM to the ATP-activated RyR2 channels. Notably, this concentration range encompassed the therapeutically relevant levels of 2–5 µM [45]. Figure 1A shows representative current traces of the RyR2 channel at 0 mV before and after the Tris^+^ gradient was generated and in response to 5, 50, and 200 µM flecainide. Previous studies have concluded that flecainide acted as a partial blocker of the RyR2 conducting pore [26,27,28,29]; however, we did not observe any partial closures to a subconductance state as evidence of partial occlusion of the channel pore. All-point amplitude histograms shown alongside the current traces were fitted with the sum of two Gaussian functions corresponding to the main open and closed states (Figure 1A).

To further evaluate the blocking potency of flecainide, we analyzed the dose response of RyR2 G for flecainide. Figure 1B clearly shows that establishing the 125/10 mM Tris^+^ gradient resulted in a noticeable decreased RyR2 G (Table 1). This occurred because the generated Tris^+^ countercurrent substantially reduced the net cation flow from SR lumen to cytosol. When normalized, the G value was decreased to 79.5 ± 1.4%. Notably, the addition of flecainide gradually increased the RyR2 G, with 2 µM causing a significant rise (79.5 ± 1.4% vs. 87.8 ± 2.1%). At concentrations of 10 µM and higher, the RyR2 G approached values similar to those observed when the Tris^+^ gradient was eliminated, and, thus, the Tris^+^ countercurrent was reduced to zero. These findings are consistent with a scheme in which flecainide enters the RyR2 conducting pathway and blocks the Tris^+^ countercurrent. Hence, the block appears as a time-averaged reduction in the values of RyR2 G, flecainide likely rapidly dissociates from its binding site within the channel pore. It is not an unreasonable proposal because Figure 1C demonstrates that flecainide has no significant impact on the values of RyR2 P_O_ across the entire range of tested concentrations, which is characteristic behavior commonly observed for very fast channel blockers [37]. Furthermore, the lack of significant differences in the dependence of RyR2 P_O_ on flecainide concentration supports the widely accepted view that the RyR2 channel does not respond to flecainide when the net cation current through the channel flows from SR lumen to cytosol [28,29].

We extended the investigation presented above by examining the blocking effect of flecainide on a larger Tris^+^ countercurrent driven by a 250/10 mM Tris^+^ gradient. Representative current traces of the RyR2 channel recorded at 0 mV, and the corresponding all-points amplitude histograms in Figure 1D again show no evidence of partial closings indicative of partial occlusion of the RyR2 pore by flecainide. The channel simply shuttled between the main open state and closed state. The increase in Tris^+^ concentration from 10 mM to 250 mM led to a substantial decrease in RyR2 G (Table 1). When normalized, the G value decreased to 72.4 ± 1.8%. Starting from 5 µM flecainide, the RyR2 G was significantly increased (72.4 ± 1.8%. vs. 81.5 ± 2.6%), and only at 200 µM flecainide, it approached the value determined in the absence of a Tris^+^ gradient (Figure 1E). Furthermore, Figure 1F demonstrates that RyR2 P_O_ was not affected by flecainide, similar to the response observed for the 2-fold smaller Tris^+^ gradient (Figure 1C). The observations mentioned above for the 250/10 mM Tris^+^ gradient further support the proposal that flecainide acts as a very fast blocker of the RyR2 channel under the experimental conditions tested in our study.

Since flecainide was dissolved in DMSO, we performed a set of experiments to exclude potential interference from this solvent. For both 125/10 mM and 250/10 mM Tris^+^ gradients, only DMSO (0.0067–0.667% (*v*/*v*)), in an amount equivalent to that used when flecainide was tested, was applied to the cytosolic side of the RyR2 channel. Under both conditions, the values of RyR2 G remained unchanged with increasing DMSO concentration (Appendix A). Similarly, the RyR2 P_O_ was not affected (Appendix A). These results clearly indicate that the observed changes in the RyR2 G were solely attributable to the action of flecainide.

The experiments reported up to this point have been conducted when the Tris^+^ countercurrent was present. To clearly demonstrate that the specific target of flecainide is the RyR2-mediated countercurrent, we performed the same set of experiments with the Tris^+^ driving force reduced to zero (10/10 mM Tris^+^). The results collected are summarized in Figure 2. Representative current traces of the RyR2 channel recorded in response to 5, 50, and 200 µM flecainide show no evidence of partial closures (Figure 2A). The corresponding all-point amplitude histograms displayed only two peaks, representing the main open state and closed state. As expected, the values of RyR2 G did not change appreciably with increasing flecainide concentration (Figure 2B). Similarly, the RyR2 P_O_ was not altered by flecainide (Figure 2C).

To quantitatively compare flecainide’s ability to block the RyR2-mediated countercurrent driven by Tris^+^ gradients of two different magnitudes, the G values, plotted as a function of flecainide concentration, were fitted by the Hill equation (Figure 3A). The solid lines derived for the 125/10 mM and 250/10 mM Tris^+^ gradients represent the best global fitting curves. Figure 3B shows that the fitted EC_50_ for flecainide was not significantly affected by the magnitude of the Tris^+^ gradient (1.8 ± 1.4 µM for 125/10 mM Tris^+^ vs. 13 ± 13 µM for 250/10 mM Tris^+^). Although there was a visible trend indicating a decrease in flecainide`s potency when a larger Tris^+^ countercurrent flowed through the RyR2 channel, the difference in the EC_50_ was not statistically significant. Under both tested conditions, the fitted values of G^max^ were similar (98.0 ± 7.1% for 125/10 mM Tris^+^ vs. 98.6 ± 8.5% for 250/10 mM Tris^+^). This result, combined with the finding that 200 µM flecainide restored the RyR2 G values to the control level (no presence of Tris^+^ countercurrent and flecainide) (Figure 1B,E), strongly indicates that flecainide, regardless of the magnitude of the Tris^+^ gradient, is capable of completely blocking the countercurrent through the RyR2 channel as its concentration approaches 200 µM. Finally, a significant change was only observed in the fitted value of G_0_ corresponding to the G value in the absence of flecainide (78.9 ± 1.9% for 125/10 mM Tris^+^ vs. 72.5 ± 1.6% for 250/10 mM Tris^+^). With respect to the absolute values, the G decreased from 172.3 ± 4.2 pS to 147.1 ± 3.8 pS when the Tris^+^ gradient was increased (Table 1). This was expected, as doubling the driving force for the Tris^+^ countercurrent should result in a greater decrease in RyR2 G unless the Tris^+^ countercurrent had already been saturated. The obtained results clearly demonstrate that saturation did not occur. In the absence of a Tris^+^ gradient, the G versus flecainide relationship exhibited a very weak correlation, as evidenced by a Pearson correlation coefficient of r = −0.221. This indicates that there was no consistent linear association between the RyR2 G and flecainide, further supporting the lack of flecainide’s effect on RyR2 G (Figure 2B) when only the Ca^2+^/Ba^2+^ current flowed through the channel in the SR lumen-to-cytosol direction. Overall, it is most likely that flecainide’s target is the RyR2-mediated countercurrent, and its blockage is highly specific under cell-like conditions where the net cation current in the SR lumen-to-cytosol direction predominates.

### 2.2. Effect of Flecainide on the Function of SR Cl^−^ Channels

Mathematical simulations suggest that the SR countercurrent during Ca^2+^ release in cardiac muscle cells is a multifactorial process involving several potential pathways that together form a functional network [46]. One potential component of this complex countercurrent system is SR Cl^−^ channels. We, therefore, investigated the potential action of flecainide on these channels, as our long-term experience with single-channel measurements suggests they are abundant in SR microsomes isolated from the rat heart. Since it has been demonstrated that RyR2 channels incorporate into BLMs with the cytosolic side facing the cis compartment [44], and the SR Cl^−^ channels often incorporate into BLMs in the same fusion events as RyR2 channels, it is reasonable to assume that the SR Cl^−^ channels will adopt a similar orientation. Therefore, flecainide was added only to the cis solution. Cardiac SR microsomes also contain K^+^ channels, and as we used KCl to generate the driving force for Cl^−^ flow through SR Cl^−^ channels, we selected the reversal potential (E_rev_) as a parameter to clearly distinguish between these channels since each channel type exhibits E_rev_ of opposite polarity. Previous studies have established, and our work further extended, that the blocking action of flecainide is dependent on the RyR2 current direction [28,29]. It was, thus, reasonable to test the potential effect of flecainide on the Cl^−^ current flowing through SR Cl^−^ channels in both directions. Figure 4A shows representative current traces of the SR Cl^−^ channel before and after the cis addition of 5, 50, and 200 µM flecainide. In this case, the Cl^−^ current flowed from cytosol to the SR lumen driven by the 450/150 mM Cl^−^ gradient (E_rev_ = −25.1 ± 1.7 mV). In contrast, Figure 4D displays representative current traces when the direction of the Cl^−^ current was reversed by inverting the Cl^−^ gradient (150/450 mM Cl^−^, E_rev_ = +25.0 ± 1.2 mV). Under both tested conditions, we did not observe any partial closings indicative of partial channel blockage. The all-points amplitude histograms display only one major peak corresponding to the main open state, with a small or negligible peak corresponding to the closed state. SR Cl^−^ channels were almost fully active (the control absolute value of P_O_ is 0.936 ± 0.023 for 450/150 mM Cl^−^ and 0.944 ± 0.015 for 150/450 mM Cl^−^) even under diastolic conditions ([Ca^2+^]_C_~90 nM, [Ca^2+^]_L_ = 1 mM) without addition of any known activator, and, thus, remained in the main open state for most of the time. Irrespective of the direction of Cl^−^ flow, SR Cl^−^ channels exhibited a similar G (Table 1), and flecainide produced no discernible effect on this parameter (Figure 4B,E). Neither was the P_O_ affected by flecainide under both tested conditions (Figure 4C,F). Taken together, our results indicate that SR Cl^−^ channels are not physiologically relevant targets of flecainide, and thus, the countercurrent potentially carried by these channels is unlikely to contribute to the therapeutic antiarrhythmic effect of flecainide in CPVT treatment.

### 2.3. Interaction of Flecainide with the Lipid Membrane Evaluated by Quartz Crystal Microbalance (QCM) Measurement

Although 99% of flecainide exists in the protonated form at physiological pH [18], it is thought to specifically interact with the lipid membrane [47], consisting of a lipophilic aromatic ring structure and a protonated piperidine ring. This interaction may enable flecainide to reach its binding site within the pore of the voltage-gated Na^+^ channel from the membrane environment, utilizing hydrophobic pathways known as fenestrations [36]. Since fenestrations have been suggested to have a pharmacological significance for a broader spectrum of ion channels, we decided to assess whether flecainide interacts with the lipid membrane under our experimental conditions. We employed the QCM method, which allows for label-free monitoring of events occurring at the sensor surface. A lipid membrane can be formed over the SiO_2_ surface by immobilizing liposomes (lipid vesicles), thereby enabling the study of lipid membrane interactions with a variety of molecules [48,49]. A decrease in the frequency at which the QCM sensor oscillates occurs when molecules specifically bind to the surface, and this change is proportional to the amount of mass bound [38,39,40,41,50].

To form a lipid membrane on the SiO_2_ surface of the QCM sensor, we used a liposome suspension prepared from a mixture of DOPE and DOPC, similar to that utilized in the single-channel experiments. Figure 5A demonstrates the formation of the lipid membrane on the SiO_2_ surface when the liposome suspension was perfused through the flow cell. Initially, spontaneous liposome adsorption on the sensor surface occurred, which was indicated by a fast decrease in Δf_S_. The subsequent formation of the lipid membrane, driven by the fusion of neighboring liposomes, was facilitated by the addition of 5 mM CaCl_2_. This resulted in a rapid small increase in Δf_S_. After washing with the cis solution, the total decrease in Δf_S_ was −45.5 ± 6.0 Hz. Once the frequency was stabilized, flecainide was added.

The frequency changes as a function of flecainide concentration ranging from 0.2 µM to 10 µM are shown in Figure 5B. Each addition significantly increased the −Δf_S_ from 4.88 ± 0.66 Hz for 0.2 µM flecainide to 19.20 ± 0.96 Hz for 10 µM flecainide, indicating a notable interaction of flecainide with the lipid membrane. In Figure 5D, the −Δf_S_ plotted as a function of flecainide concentration was fitted with the Langmuir adsorption isotherm to quantify the extent of flecainide binding. The fitted value of ${-\Delta f}_{S}^{max}$ = 18.9 ± 2.8 Hz implies that the saturation was reached at 10 µM flecainide, which is fully consistent with the observation obtained for the RyR2 G versus flecainide relationship when the 125/10 mM Tris^+^ gradient was utilized (Figure 1B and Figure 3A). The fitted value of K_L_ = 1.55 ± 0.18 × 10^6^ M^−1^ indicates that the interaction of flecainide with the lipid membrane was substantially strong.

DMSO, which was used to dissolve flecainide, is known to interact with the lipid membrane and potentially cause its disintegration [51,52,53]. Although the DMSO concentrations corresponding to 0.2–10 µM flecainide were very low (ranging from 0.0007% to 0.033% (*v*/*v*)), we performed a set of experiments with DMSO alone to ensure that the solvent did not contribute to the changes caused by the addition of flecainide. After the lipid membrane was formed on a SiO_2_ surface of the QCM sensor, DMSO was added stepwise (Appendix A). Appendix A clearly shows that the values of −Δf_S_ remained unchanged with increasing DMSO concentration, which supports the finding that DMSO adsorbs onto the membrane surface by displacing water molecules only at concentrations exceeding 0.39% (*v*/*v*) [54,55]. This is more than 10 times higher than that achieved in our QCM study. In addition, we demonstrated that the Δf_S_ signal remains highly stable over time. We can, therefore, conclude that the observed changes in Δf_S_ were attributable to flecainide and not affected by DMSO or temporal fluctuations.

To confirm the specificity of flecainide binding to the lipid membrane, caffeine was tested under the same experimental conditions (Figure 5C). Caffeine is soluble in both water and polar organic solvents but is significantly less soluble in non-polar solvents such as lipids [56]. Therefore, we expected little to no interaction of caffeine with the lipid membrane. When lower concentrations of caffeine (0.2, 0.5, and 1 µM) were injected, a slight but not significant decrease in frequency was observed (−1.40 ± 0.20 Hz for 0.2 μM flecainide and −3.19 Hz for 1 μM flecainide). In line with this, the −Δf_S_ versus flecainide relationship showed a weak correlation, as indicated by a Pearson correlation coefficient of r = 0.372 (Figure 5D). When higher concentrations of caffeine (2, 5, and 10 µM) were tested, lipid membrane washout likely occurred, as only an increase in Δf_S_ was observed, indicating a loss of mass bound to the sensor surface. The membrane–caffeine interaction can, thus, be considered very weak, given the low solubility of caffeine in non-polar solvents [56]. Overall, our observations strongly indicate that the decrease in Δf_S_ upon interaction with flecainide is highly likely attributable to its specific binding to the lipid membrane.

## 3. Discussion

Given the importance of flecainide in the treatment of CPVT, numerous studies have sought to understand the molecular mechanisms underlying its therapeutic action (reviewed in [20,25]). It has been proposed that flecainide reduced proarrhythmic behavior through the combined effects on Ca^2+^ and Na^+^ signaling, involving the blockage of both the plasma membrane Na^+^ channel and the intracellular RyR2 channel in cardiac muscle cells. Which mechanism has greater dominance in eliciting the antiarrhythmic effect of flecainide remains unclear; however, it may vary among patients and depend on the clinical state of CPVT [20,23]. While the blocking effect of flecainide on the voltage-gated Na^+^ channel (Nav1.5) is generally well established [17,18,57,58], the blockage of the RyR2 channel by flecainide has been the subject of extensive debate, particularly concerning its physiological significance (as reviewed in [25]). A key point has been that flecainide only blocks the RyR2 current in the opposite direction to that experienced during Ca^2+^ release from the SR. A recent paper by Bannister et al. [20] aimed to address this controversy, proposing a hypothesis that a component of flecainide’s therapeutic action involves a partial blockage of the RyR2-mediated countercurrent (from cytosol to SR lumen), which has also been suggested to have physiological relevance as a charge-compensating current needed to support a massive Ca^2+^ release from the SR [46,59]. Thus, it is not an unreasonable proposal that impairing the charge compensation may indeed result in the inhibition of Ca^2+^ release [28,29,30,34]. In this study, we set out to validate the hypothesis posited by Bannister et al. [20] by recording the activity of RyR2 channels reconstituted into BLMs under cell-like conditions.

### 3.1. The RyR2-Mediated Countercurrent as an Intracellular Target of Flecainide

Our data demonstrate that flecainide can specifically block the RyR2-mediated countercurrent driven by a Tris^+^ gradient, even when a dominant Ca^2+^/Ba^2+^ current flows in the opposite direction. We observed that 200 µM flecainide completely attenuated the Tris^+^ countercurrent at two distinct amplitudes, demonstrating consistent efficacy of flecainide under both conditions (Figure 1B,E). This occurred at a concentration well above the therapeutic range of 2–5 µM [45]. However, even at clinically relevant concentrations, we observed a significant increase in RyR2 G resulting from the reduction in Tris^+^ countercurrent at both tested amplitudes, strongly suggesting an interaction between flecainide and the RyR2 conducting pathway [28]. Although our data did not show a statistically significant correlation between the blocking potency of flecainide and the amplitude of Tris^+^ countercurrent, a substantial positive shift in the EC_50_ for flecainide, beyond clinically relevant concentrations, was apparent when the magnitude of the Tris^+^ gradient was doubled. This suggests that flecainide is less potent at blocking the Tris^+^ countercurrent with higher amplitudes (Figure 3). Notably, this finding is fully consistent with the observation of Mehra et al. [27] that the IC_50_ for flecainide (derived from fitting the RyR2 Po versus flecainide relationships) was noticeably lower when the driving force for Cs^+^ flow through the RyR2 channel in the cytosol-to-SR lumen was reduced. Recently, Steer et al. [35] reported that when SR K^+^ and Cl^−^ channels, a relevant source of SR countercurrents, were inhibited in permeabilized cardiomyocytes, flecainide’s effect on SR Ca^2+^ wave frequency was potentiated. Since charge compensation is essential for a robust SR Ca^2+^ release [30,46], we propose that the RyR2-mediated countercurrent, as the only remaining source of charge-compensating currents, must have increased under these conditions, thereby enhancing the effect of flecainide. Collectively, these results strongly indicate that the flecainide binding within the cytosolic portion of the RyR2 conducting pore might be biphasically regulated by the amplitude of channel-mediated countercurrent. This new aspect of flecainide’s action requires further investigation because the amplitude of the RyR2-mediated countercurrent may differ between healthy individuals and CPVT patients.

In contrast to the aforementioned arguments for the RyR2-mediated countercurrent, Bannister et al. [28] suggested that flecainide binding was negatively modulated by the RyR2 current in the opposite direction, as the interaction was not strong enough to prevent flecainide displacement caused by cations flowing in the SR lumen-to-cytosol direction. However, our findings apparently contradicted this reasoning, as the blocking effect of flecainide on the Tris^+^ countercurrent was clearly evident, even though a dominant reverse Ca^2+^/Ba^2+^ current flowed through the RyR2 channel simultaneously (Figure 1B,E). Recently, Bannister et al. [20] proposed the view that the coexistence of opposing current and countercurrent could favor the flecainide interaction in the RyR2 conducting pore, and our results clearly validate this concept. Furthermore, this also explains why others reported no effect of flecainide when only cation flow in the SR lumen-to-cytosol direction was present [28,29].

Based on multiple published results, it has been established that flecainide was an open-state, partial blocker of the RyR2 channel, causing only partial occlusion of the RyR2 conducting pore [22,23,26,27,28,29]. This action is manifested by the occurrence of partial closings to the subconductance state and a reduction in RyR2 Po. Under our experimental conditions, however, such changes were not observed (Figure 1). Aside from that, our data show a noticeable concentration-dependent effect on RyR2 G, which likely arose from a very short duration of elementary blocking events caused by flecainide binding. As a result, individual blocking events were not resolved, and thus, only a global change in RyR2 G was observed [37]. Here, we can speculate that the simultaneous flow of opposing current and countercurrent through the RyR2 channel enables a specific mode of flecainide binding in the RyR2 pore, manifested by a significant increase in the frequency of short-lived blocking events. Thus, we conclude that flecainide acts as a concentration-dependent, very fast blocker of the RyR2-mediated countercurrent when opposing current in the SR lumen-to-cytosol direction is also present.

It is widely accepted that the SR countercurrent is essential for balancing the loss of positive charges from the SR during Ca^2+^ release. Despite decades of effort [60,61,62,63,64,65], the exact nature of these countercurrents has not yet been resolved. Mathematical simulations, providing valuable insight into this field, have proposed that the SR countercurrent is not mediated by a single system but rather involves the engagement of all possible sources, including K^+^, Cl^−^, and RyR2 channels [46]. This also implies their mutual substitutability, such that even when one system is compromised, other systems can compensate to ensure robust SR Ca^2+^ release, which is fundamental for rhythmic heart contraction. Considering these theoretical predictions, we also tested the blocking effect of flecainide on SR Cl^−^ channels, which are proposed to contribute to the SR countercurrent, albeit to a smaller extent than SR K^+^ channels [46]. As with SR K^+^ channels [28], SR Cl^−^ channels exhibited resistance to flecainide, even when tested under two different experimental conditions, each with Cl^−^ flowing through the channel in only one direction (Figure 4). It appears that the sole component of the SR countercurrent responsive to flecainide is that mediated by the RyR2 channel. One might, therefore, ask, “Could blocking this particular source of SR countercurrent be sufficient to reduce aberrant SR Ca^2+^ release?” The answer greatly depends on the proximity of SR channels, as localized Ca^2+^ release is thought to require a focal countercurrent [46]. Although the exact location of SR K^+^ and Cl^−^ channels relative to RyR2 channels is unknown, mathematical simulations have shown that SR K^+^ channels would account for ~99% of the K^+^ countercurrent when positioned near RyR2 channels. However, if SR K^+^ channels are considerably distant from RyR2 channels, a significant portion of the K^+^ countercurrent (~30%) would be mediated by RyR2 channels [46]. Given this prediction and the fact that at clinically relevant concentrations (2‒5 µM), flecainide was able to cause a moderate attenuation of the Tris^+^ countercurrent by ~50%, it is reasonable to suggest that blocking the RyR2-mediated countercurrent might have significant physiological relevance.

### 3.2. Is Hydrophobic Pathway Involved in the Entry of Flecainide to the RyR2 Conducting Pore?

In the voltage-gated Na^+^ channel, flecainide binds in the central cavity of the channel pore [36], causing three different state-dependent modes of blockage [19,58,66,67]. According to the modulated receptor model proposed by Hille [37], the resting closed-state block occurs when the drug reaches its binding site by diffusing through the membrane lipid phase (hydrophobic pathway), while the rapid open-state block takes place as the drug enters the open pore from the cytosol (hydrophilic pathway). For numerous voltage-gated Na^+^ and K^+^ channels, specific hydrophobic pathways connecting the membrane to the central cavity, known as fenestrations, have been visualized in 3D structures [36,68,69,70,71,72]. However, only recently have fenestrations been shown to provide an access pathway for resting closed-state blocks of the voltage-gated Na^+^ channel by flecainide [36]. Since the pharmacological significance of fenestrations has been proposed as a general phenomenon [36], it may also be extended to the interaction of the RyR2 channel with flecainide. At physiological pH (7.35) used in our study, 99% of flecainide exists in the protonated form [18]. Therefore, one would expect preferential binding of flecainide in the water–lipid membrane interface. This behavior has been experimentally demonstrated for various local anesthetics [73], which render their therapeutic effects via blockage of the voltage-gated Na^+^ channels, similar to class I antiarrhythmic drugs [37,74,75,76]. Employing the QCM method, we indeed found that flecainide interacted with the lipid membrane supported on a SiO_2_ surface of the QCM sensor, using the same experimental conditions as for single-channel experiments (Figure 5). Although the QCM method is not sufficient to determine whether flecainide penetrates deeper into the lipid membrane or remains on its surface, mathematical simulations resulted in predictions that flecainide would lose its charge on the membrane surface, facilitating its partitioning into the hydrophobic region [47]. Thus, if the RyR2 channel possesses a fenestration region, it is plausible to suggest that in our experiments, flecainide might have accessed its binding site within the channel pore through the membrane environment. Our observations that the binding of flecainide on the lipid membrane is dose-dependent and well-correlated with the blocking effect on the RyR2-mediated countercurrent (Figure 1 and Figure 5) are consistent with this expectation. On the other hand, our suggestion does not receive support from the modulated receptor model [37], which associates the hydrophobic pathway with the resting closed-state block that has a slow blocking mechanism. In our study, however, flecainide appeared to act as a very fast blocker. Notably, in addition to the predominant fast blockage of the RyR2 channel by flecainide, Mehra et al. [27] also identified a slow blocking mode, which may be mediated by a hydrophobic access route. It is evident that over 15 years of studies on flecainide-RyR2 interactions have not provided a clear and consistent picture of the mechanism of flecainide’s action, which obviously depends on the experimental conditions employed in single-channel studies. Many questions remain, posing a challenge for further research to analyze and dissect this complex blocking mechanism in sufficient detail.

## 4. Materials and Methods

### 4.1. Single-Channel Recordings

The sarcoplasmic reticulum (SR) microsomes enriched in RyR2 and Cl^−^channels were isolated from rat ventricular muscle as previously described in [77]. Individual activities of RyR2 and SR Cl^−^ channels incorporated into planar lipid membranes (BLMs) were recorded in separate experiments under voltage-clamp conditions. The BLMs of a 3:1 mixture (*wt*/*wt*) of 1,2-dioleoyl-sn-glycero-3-phosphoethanolamine (DOPE) and 1,2-dioleoyl-sn-glycero-3-phosphocholine (DOPC) were painted on 50–70 µm diameter circular apertures in the wall of a polystyrene cup, separating the cis and trans compartments. RyR2 channels incorporate into the BLMs in a fixed orientation, such that the cis compartment corresponds to the cytosol and the trans compartment corresponds to the lumen [44]. Since Cl^−^ channels are located in the SR microsomes together with RyR2 channels, it is reasonable to assume that the cytosolic side of Cl^−^ channels will also face the cis compartment. For RyR2 channels, the cis compartment was filled with 1 mL of 150 mM KCl and either 10, 125, or 250 mM Tris. The pH was adjusted to 7.35 using 20, 250, or 500 mM HEPES, respectively. The free cytosolic Ca^2+^ concentration ([Ca^2+^]_C_) of 90 nM was obtained by including 1 mM ethylene glycol-bis(2-aminoethylether)–N,N,N′,N′-tetraacetic acid (EGTA) and 0.480–0.587 mM CaCl_2_, depending on the ionic strength manipulated by varying the concentrations of Tris and HEPES. Free [Ca^2+^]_C_ and the ionic strength were calculated by WinMaxc32 version 2.50 (Chris Patton, Stanford University, CA, USA). The trans compartment was filled with 1 mL of 1 mM Ca(OH)_2_, 20 mM Ba(OH)_2_, 150 mM KCl, 10 mM Tris, and 140 mM HEPES (pH = 7.35). The luminal concentration of Ca^2+^ ([Ca^2+^]_L_) was maintained at the physiological level. Before applying 2‒200 µM flecainide (an acetate salt) to the cytosolic RyR2 face, 1‒3 mM ATPNa_2_ was added for moderate activation of the channel. For SR Cl^−^ channels, the cis compartment was filled with 1 mL of 450 mM KCl, 10 mM Tris, and 20 mM HEPES (pH = 7.35). The free [Ca^2+^]_C_ of 90 nM was achieved by mixing 1 mM EGTA with 0.480 mM CaCl_2_. The trans compartment was filled with 1 mL of 1 mM Ca(OH)_2_, 150 mM KCl, 10 mM Tris, and 25 mM HEPES (pH = 7.35). Under these conditions, the Cl^−^ current flowed from the cis to the trans compartment. For testing the effect of flecainide when the Cl^−^ current flow was in the opposite direction, the cis compartment was filled with 1 mL of 150 mM KCl, 10 mM Tris, and 20 mM HEPES (pH = 7.35). The free [Ca^2+^]_C_ of 90 nM was achieved by mixing 1 mM EGTA with 0.587 mM CaCl_2_. The trans compartment was filled with 1 mL of 1 mM Ca(OH)_2_, 450 mM KCl, 10 mM Tris, and 25 mM HEPES (pH = 7.35). Under both tested conditions, the SR Cl^−^ channels were almost fully active; therefore, no additional activator was added before applying flecainide. After incorporating either RyR2 or Cl^−^ channels into the BLM, a 10 mM flecainide solution was prepared by diluting a 30 mM stock solution (in DMSO) with the corresponding cis solution. The aqueous solution was not stored for longer than one day. Flecainide to final concentrations of 2, 5, 10, 20, 50, 100, and 200 µM was added to the cytosolic side of RyR2 and SR Cl^−^ channels, which were exposed to each flecainide concentration for more than 3 min. To exclude interference from DMSO, only this solvent was added to the cytosolic side of the RyR2 channel when the Tris^+^ gradient was present. DMSO to final concentrations of 0.0067, 0.166, 0.333, and 0.667% (*v*/*v*), corresponding to flecainide of 2, 50, 100, and 200 µM, were tested, respectively. The trans compartment was connected to the head-stage input of a Warner BC-535D amplifier (Warner Instruments, Inc., Hamden, CT, USA), and the cis compartment was held at the ground level. Electrical signals were filtered through the Warner BC-535D low-pass Bessel filter at 1 kHz and digitized at 4 kHz with an A/D-D/A converter (Digidata 1322 A, Molecular Devices, Sunnyvale, CA, USA).

### 4.2. Quartz Crystal Microbalance (QCM) Measurement

QCM sensor preparation: Prior to the experimental procedure, the QCM sensor with a silicon dioxide (SiO_2_) layer for liquid biosensing (QCM 5 MHz 14 mm Cr/Au/SiO_2_ Q Sense, QuartzPro, Jarfalla, Sweden), having a fundamental frequency of 5 MHz, was subjected to cleaning in a basic piranha solution (30% hydrogen peroxide: 96% ammonia: water, 1:1:5) heated to 70 °C for 25 min, followed by rinsing with water. This procedure was repeated three times and finished by drying the sensor with a stream of nitrogen gas. The cleaned sensor was placed in the flow cell of a QCM circuit.

Liposome preparation: DOPC (transition temperature T_C_ = −20 °C) and DOPE (T_C_ = −16 °C), each dissolved in chloroform at a concentration of 10 mg/mL, were mixed at a 1:1 weight ratio. A volume of 150 µL from each lipid solution was added to a round-bottom flask and thoroughly vortexed. The chloroform was evaporated to dryness using a vacuum pump (Merck Millipore, Darmstadt, Germany). The resulting lipid film was hydrated with 3 mL of the cis solution at room temperature and agitated for 20 min to obtain a liposome suspension. The liposomes were allowed to stand overnight at room temperature to facilitate the aging process, which improved size homogeneity and simplified the sizing process. Prior to each experiment, the liposome suspension was sonicated for 10 min in a bath sonicator. A fresh liposome suspension was prepared for each QCM measurement.

QCM measurement: In the first step, the cis solution containing 20 mM HEPES, 10 mM Tris, 150 mM KCl, 1 mM EGTA, and 0.587 mM CaCl_2_ ([Ca^2+^]_free_ = 90 nM) was perfused through the flow cell until a stable frequency (f_S_) was observed. A suspension of liposomes was then perfused into the flow cell for 20 min to allow for spontaneous immobilization of liposomes onto the sensor surface. To facilitate the formation of a planar lipid membrane, 5 mM CaCl_2_ was added to the cis solution and perfused into the flow cell for five min, promoting liposome fusion, followed by rinsing with the cis solution. When the lipid membrane was spread over the surface of the QCM sensor, a solution of either flecainide or caffeine was applied for 30 min, and the frequency change (Δf_S_) was monitored for each addition. The same set of experiments was performed with DMSO alone to establish a solvent control and to test temporal stability. DMSO at final concentrations of 0.0007, 0.0017, 0.0034, 0.0067, 0.017, and 0.033% (*v*/*v*), corresponding to flecainide concentrations of 0.2, 0.5, 1, 2, 5, and 10 µM, respectively, was examined. Values of Δf_S_ were calculated as the average of recorded points obtained during the final minute of each addition. In total, 600 points were averaged for each concentration.

### 4.3. Drugs and Chemicals

Phospholipids were obtained from Avanti Research, Inc. (Alabaster, AL, USA). Chemicals for single-channel recordings with the highest available purity were from Sigma-Aldrich (St. Louis, MO, USA). Chemicals for QCM measurements with the highest available purity were from Centralchem (Bratislava, Slovakia). Ultrapure water with a resistivity of 18.5 MΩ.cm at 25 °C (Simplicity 185, Merck Millipore, Darmstadt, Germany) was used to prepare all solutions.

### 4.4. Single-Channel Data Acquisition and Analysis

Data acquisition and analysis were performed with a commercially available software package (pCLAMP 10.5, Molecular Devices, Sunnyvale, CA, USA). The open probability (P_O_), a measure of channel activity, was calculated from continuous records of >2 min in duration, collected at 0 mV, using the 50%-amplitude threshold method. The conductance (G) and the reversal potential (E_rev_) were determined from a linear regression of the current–voltage relationship, acquired by applying membrane voltage sequentially from −10 mV to +20 mV in 5 mV increments. For each membrane voltage, a continuous recording of approximately 10 s in duration was made. The slope of the fitted line was equal to G, and E_rev_ was taken as the intersection of the voltage axis with the linear fit. Current amplitude calculations were made by measuring the difference between the means of two Gaussian curves fitted to all-point amplitude histograms. To minimize the impact of dispersion in the absolute values of G and P_O_, the flecainide-dependences of G and P_O_ are plotted in a dose-normalized manner. To compare the flecainide’s action on RyR2 G in relation to the Tris^+^ gradient across the BLM, the dependence of G on flecainide concentration was globally fitted by the Hill function,
(1)$$G=G_{0}+(G^{max}-G_{0})\frac{{\left[Flecainide\right]}^{n_{H}}}{{\left[Flecainide\right]}^{n_{H}}+{{EC}_{50}}^{n_{H}}}$$
where G_0_ is the G in the absence of flecainide; G^max^ is the maximal achievable G in the presence of flecainide, and EC_50_ is the concentration of flecainide that produces 50% of the maximum response. When the Tris^+^ gradient was absent, the dataset for G was fitted with a linear function, and the strength of the relationship between the flecainide concentrations and the values of G was measured by Pearson correlation coefficient (r).

### 4.5. QCM Data Acquisition and Analysis

The values of Δf_S_ were acquired by Maxtek RQCM (Inficon, Bad Ragaz, Switzerland) operated with RQCM Data Logging Software 2.0.3 and connected to a syringe pump (Genie Plus, Kent Scientific, Torrington, CT, USA). The dependence of −Δf_S_ on flecainide concentration was globally fitted by the Langmuir adsorption isotherm,
(2)$${-\Delta f}_{S}=\frac{-{\Delta f}_{S}^{max}K_{L}[Fleainide]}{1+K_{L}[Flecainide]}$$
where the ${-\Delta f}_{S}^{max}$ is related to the maximal adsorption capacity, and K_L_ is the Langmuir constant, a measure of how strongly flecainide interacts with the lipid membrane on the QCM sensor surface. When caffeine was tested, the dataset for −Δf_S_ was fitted with a linear function, and the strength of the relationship between the flecainide concentrations and the values of −Δf_S_ was measured by Pearson correlation coefficient (r).

### 4.6. Statistical Analysis

The results are reported as the average ± SEM or the fitted value ± SEM. For DMSO testing, statistical comparison of differences in the values of G, P_O_, and −Δf_S_ was made by one-way repeated measures ANOVA with Tukey’s post hoc test. All key requirements for this statistical test were met. The values of G, P_O_, and −Δf_S_ followed a normal distribution (Shapiro–Wilk test), more than one DMSO concentration was tested, no outliers were identified (based on the z-score), sphericity was tested and found to be acceptable (Mauchly’s test), and a balanced number of repeated measurements was maintained. For flecainide and caffeine testing, a mixed-effects analysis with Tukey’s post hoc test was used, providing the flexibility to manage missing data [78]. All key assumptions of the linear mixed-effects model, including normality of residuals, homoscedasticity, and normality of random effects, were evaluated and found to be satisfactory. The normality of residuals and random effects was assessed using Q–Q plots, while homoscedasticity was examined by plotting residuals against fitted values to check for constant variance. An unpaired Student’s *t*-test was performed to detect significant changes in the values of G_0_, EC_50_, and G^max^ for the flecainide-dependence of RyR2 G. The same statistical test was used to compare the absolute values of RyR2 G at 0 µM flecainide determined for each Tris^+^ gradient as well as the absolute values of G at 0 µM flecainide gained for SR Cl^−^ channels recorded under opposing Cl^−^ gradients. A Paired Student’s *t*-test was performed to detect significant changes in the values of RyR2 G at 0 µM flecainide before and after the Tris^+^ gradient was established. Differences were regarded to be statistically significant at *p* < 0.05.

## 5. Conclusions

Here, we aimed to add complementary information to establish whether flecainide was indeed capable of blocking the RyR2 channel in a physiological context. In contrast to other studies [21,22,23,26,27,28,29], we examined the blocking effect of flecainide under specific conditions when two opposing currents flowed simultaneously through the RyR2 channel. We sought to better simulate cell-like conditions when Ca^2+^ current through the RyR2 channel in the SR lumen-to-cytosol direction is highly likely accompanied by a charge-compensating current, known as countercurrent, flowing in the opposite direction. It has been proposed that blocking any of the pathways maintaining charge neutrality across the SR membrane may indeed result in the inhibition of SR Ca^2+^ release [28,29,30,34]. In line with this, we found that flecainide was a potent, very fast blocker of the RyR2-mediated countercurrent, as evidenced by changes in the RyR2 conductive properties without affecting channel activity. Notably, flecainide did not affect SR Cl^−^ channels despite their strong potential in charge balancing during Ca^2+^ release. Our findings strongly support the hypothesis posited by Bannister et al. [20] that flecainide would not be expelled from its binding site situated in the RyR2 pore when both directionally opposing currents simultaneously flowed through the channel. Overall, our observations contribute to the validation of the RyR2 channel as an intracellular therapeutic target for treating CPVT patients, thereby refocusing our attention on the molecular basis of the CPVT disorder.

## Figures and Tables

**Figure 1 ijms-26-00203-f001:**
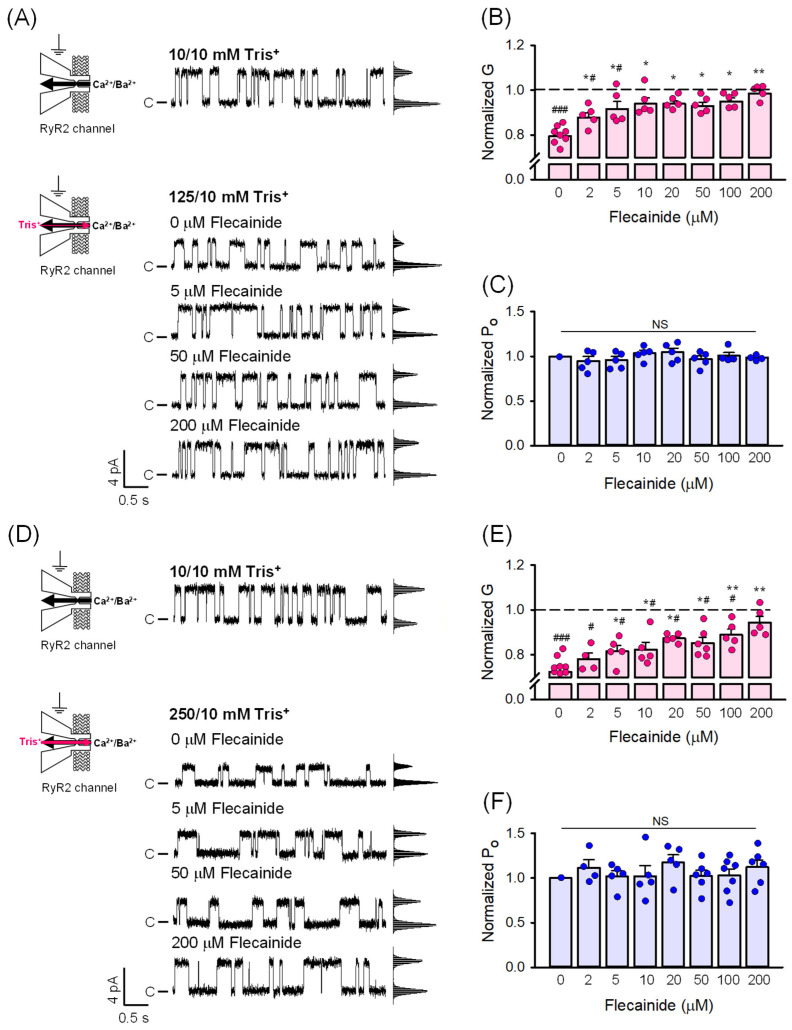
Flecainide blocks the RyR2-mediated countercurrent. Representative current traces of RyR2 channels recorded at 0 mV when equal concentrations of Tris^+^ (10/10 mM) on both sides of the BLM were present, followed by perfusing the cis compartment with a solution containing 125 mM (**A**) or 250 mM Tris^+^ (**D**) to drive the Tris^+^ countercurrent in the cytosol-to-SR lumen direction. A dominant Ca^2+^/Ba^2+^ current in the SR lumen-to-cytosol direction flowed through the RyR2 channel; therefore, openings are shown as upward deflections from the marked zero level. Flecainide of 5, 50, and 200 µM was added to the cis compartment corresponding to the cytosol. No partial blocking events were observed in the all-point amplitude histograms shown alongside the current traces. The increase in blocking events with increasing flecainide concentration is depicted in the RyR2 G versus flecainide relationship and manifests as an increase in the RyR2 G (**B**,**E**). No significant changes in Po were observed (**C**,**F**). Data for (**B**,**C**,**E**,**F**) are shown as average ± SEM from *n* = 8 independent experiments. Statistical significance at the level * *p* < 0.05 and ** *p* < 0.01 compared to 0 µM flecainide and ^#^ *p* < 0.05 and ^###^ *p* < 0.001 compared to 10/10 mM Tris^+^ (dashed line). NS indicates not significant.

**Figure 2 ijms-26-00203-f002:**
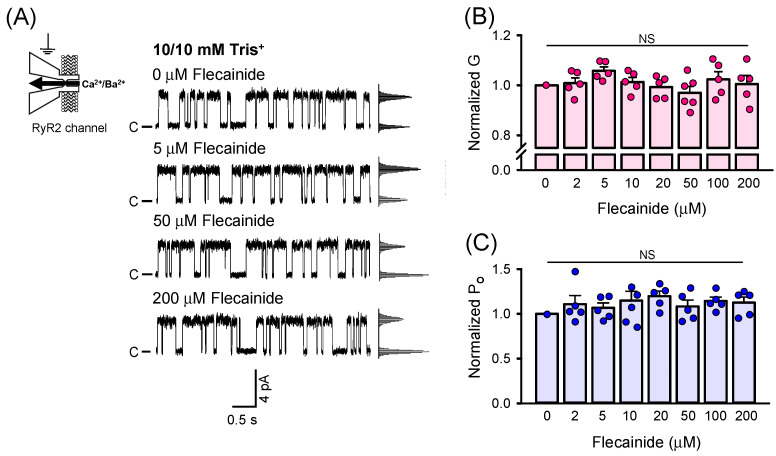
Flecainide does not affect the RyR2 channel when the Tris^+^ countercurrent is not present. (**A**) Representative current traces of the RyR2 channel recorded at 0 mV when equal concentrations of Tris^+^ (10/10 mM) on both sides of the BLM were present. A sole Ca^2+^/Ba^2+^ current in the SR lumen-to-cytosol direction flowed through the RyR2 channel; therefore, openings are shown as upward deflections from the marked zero level. Flecainide of 5, 50, and 200 µM was added to the cis compartment corresponding to the cytosol. No partial blocking events were observed in the all-point amplitude histograms, shown alongside the current traces. No significant changes in RyR2 G (**B**) and Po were detected (**C**). Data for (**B**,**C**) are shown as average ± SEM from *n* = 8 independent experiments. NS indicates not significant.

**Figure 3 ijms-26-00203-f003:**
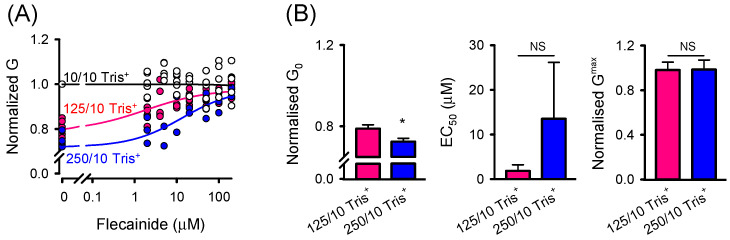
Quantitative comparison of flecainide’s ability to block the RyR2-mediated countercurrent. (**A**) The flecainide dependence of RyR2 G for 10/10 mM (white circles), 125/10 mM (red circles), and 250/10 mM (blue circles) Tris^+^ gradients. Red and blue solid lines show the best fits to the Hill equation; the black line represents the best fit to a linear function. (**B**) The fitted values of G_0_, EC_50_ for flecainide, and G^max^ obtained for 125/10 mM and 250/10 mM Tris^+^ gradients. A significant change was only observed in the fitted value of G_0_ corresponding to the G value at 0 µM flecainide. Statistical significance at the level * *p* < 0.05 compared to 125/10 mM Tris^+^ gradient. NS indicates not significant.

**Figure 4 ijms-26-00203-f004:**
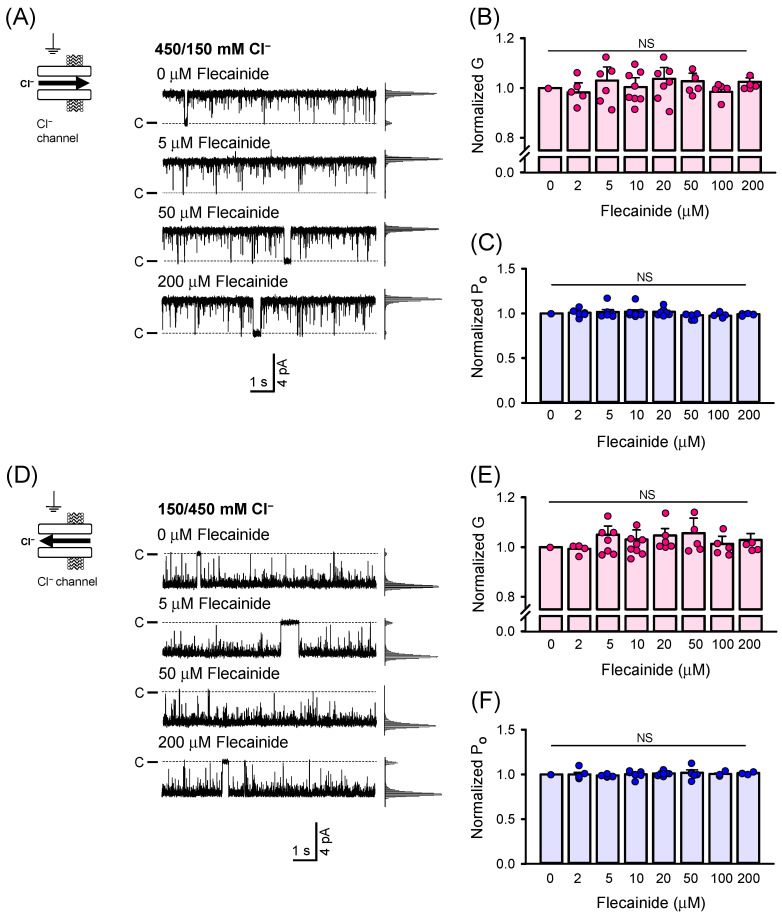
Flecainide does not target SR Cl^−^ channels. Representative current traces of SR Cl^−^ channels recorded at 0 mV when 450/150 mM Cl^−^ (**A**) or opposite 150/450 mM Cl^−^ gradient (**D**) was established. When a sole Cl^−^ current flows in the cytosol-to-SR lumen, openings are shown as upward deflections from the marked zero-level (**A**). When a sole Cl^−^ current flows in the opposite direction, openings are shown as downward deflections from the marked zero level (**D**). Flecainide of 5, 50, and 200 µM was added to the cis compartment corresponding to the cytosol. No partial blocking events were observed in the all-point amplitude histograms, shown alongside the current traces. No significant changes in G (**B**,**E**) and Po were observed (**C**,**F**). Data for (**B**,**C**,**E**,**F**) are shown as average ± SEM from *n* = 9 independent experiments. NS indicates not significant.

**Figure 5 ijms-26-00203-f005:**
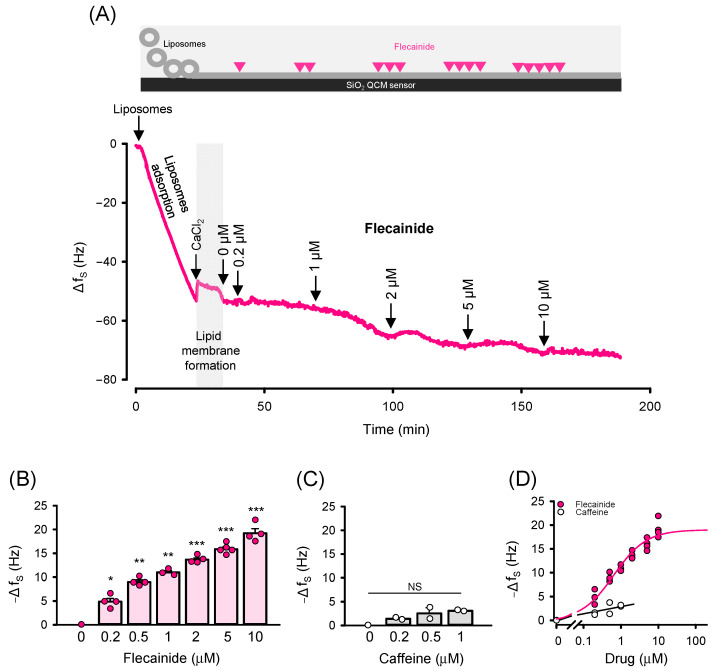
Flecainide significantly interacts with the lipid membrane supported on a SiO_2_ surface of the QCM sensor. (**A**) Representative response signal of the SiO_2_ QCM sensor when the lipid membrane was formed over the sensor surface, followed by injection of 0.2, 1, 2, 5, and 10 µM flecainide. Downward arrows indicate times of flecainide additions. The flecainide dependence of −Δf_S_ when flecainide (**B**) or caffeine (**C**) was tested. The data indicate a strong interaction between flecainide and the lipid membrane and a weak interaction with caffeine. (**D**) The −Δf_S_ plotted as a function of flecainide concentration. Red solid line shows the best fit to the Langmuir adsorption isotherm when flecainide is added; the black line represents the best fit to a linear function when caffeine is added. Data for (**B**,**C**) are shown as average ± SEM from *n* = 2–6 independent experiments. Statistical significance at the level * *p* < 0.05, ** *p* < 0.01, and *** *p* < 0.001 compared to 0 µM flecainide. NS indicates not significant.

**Table 1 ijms-26-00203-t001:** The absolute values of G for RyR2 and Cl^−^ channels in the absence of flecainide.

Type	Experimental Conditions (cis/trans)	G (pS)
RyR2 channel	10/10 mM Tris^+^	208.8 ± 6.7
	10/10 mM Tris^+^	217.24 ± 6.6
	125/10 mM Tris^+^	172.3 ± 4.2 ***
	10/10 mM Tris^+^	206.3 ± 4.8
	250/10 mM Tris^+^	147.1 ± 3.8 ***^,###^
Cl^−^ channel	450/150 mM Cl^−^	144.2 ± 8.0
	150/450 mM Cl^−^	153.9 ± 7.8

The data represent average ± SEM from *n* = 8–9 independent experiments. Statistical significance vs. corresponding 10/10 mM Tris^+^: *** *p* < 0.001, statistical significance vs. 125/10 mM Tris^+^: ^###^ *p* < 0.001.

## Data Availability

The datasets generated for this study are available upon request to the corresponding authors.

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
