# Peer review of "Flecainide Specifically Targets the Monovalent Countercurrent Through the Cardiac Ryanodine Receptor, While a Dominant Opposing Ca2+/Ba2+ Current Is Present"

_ijms, 2024, doi:10.3390/ijms26010203_

Round 1
Reviewer 1 Report
Comments and Suggestions for Authors
Gaburjakova et al have investigated the mechanisms of action of flecainide on cardiac RyR2 receptors using a single-channel recording approach. The nature of the interaction of flecainide and RyR2 receptors have been subject to debate, and this study significantly contributes to the understanding of the mechanisms. The conclusions are modest and are supported by the experimental results. The authors are congratulated for their innovative experimental approach described in this study.
Major Issue:
Is it possible that the different DMSO concentrations present when different concentrations of flecainide were added to the experimental preparations (using the stock solution) could affect the results? Are there any experiments without flecainide and only the solvent DMSO in different concentrations?
Minor Issues:
Please harmonize (use the same) colours among the different panels on Fig 3.
Reviewer 2 Report
Comments and Suggestions for Authors
Gaburjakova et al. have carried out single channel rat ventricular RyR2 bilayer recordings under conditions designed to address the direct effects of flecainide on RyR2 counter-current along with Quartz Crystal Microbalance on artificial liposomes to assess the ability of flecainide to interact directly with lipid membranes. This is an interesting topic and despite flecanide's clinical utility its precise mechanism of action clinically remains controversial, so further clarification is welcome.
The work demonstrates that the application of flecainide from the cytoplasmic face increases RyR2 conductance, which is a reflection of the reduced countercurrent. This effect is significant even at 2 uM flecainide and strongly significant by 5 uM flecainide when the Tris concentration is 125 mM:10 mM; cis:trans. These concentrations of flecainide are likely within the clinical range. However, when the driving force for the countercurrent is further increased (application of 250 mM Tris in the cis compartment relative to 10 mM in the trans), the effect of flecainide is less pronounced. No effect is observed at 2 uM, and although, a significant effect is achieved at 5 uM it is more subtle. It now takes concentrations of 20 uM and above to reach the level of significance observed at 5 uM in the 125 mM scenario.
In fig 3, although not significant, the difference in EC50 is striking. Flecainide is clearly efficacious within the clinical range for 125/10 mM Tris, however, when the driving force for the countercurrent is increased this is not the case. Why is there such dramatic variability in terms of the error of the 250/10 tris value? The IC50 value is 13 uM +/- 13 uM f. Such a large SEM suggests wildly different responses, particularly in comparison to the relatively tight errors on the 125/10 mM data. Is there a relative lack of response of any channels to flecainide in the setting of increased countercurrent? What do you think accounts for the variation of flecainide potency between the two tris gradients?
It is argued that as concentrations approach 200 uM the countercurrent is almost completely blocked irrespective of the tris gradient. This is plausibly so, but outside of the clinical range. On the basis of the above-mentioned points, is it really true that flecainide is as effective with the differing ionic gradients? It appears that flecainide is more effective when the driving force for the countercurrent is weaker, which is counterintuitive. Could the authors clarify this.
The G decreases by approximately 15% (values compared from line 233) when the tris gradient doubles. It is stated that the ‘decrease in RyR2 G should have been greater, unless the tris countercurrent was already saturated. However, this was not the case.’ What was not the case, the saturation? Or that the decrease in G was not greater?
A fairly broad spectrum of relevant literature on flecainide and RyR2 modulation has been considered yet surprisingly the work of Steer EJ et al 2023 (PMID: 37124209) appears to have been overlooked. This study investigated the effects of flecainide on RyR2 countercurrent in rat ventricular myocytes, albeit through different techniques. This should absolutely be discussed within the context of the present study approach/rationale and comparative findings.
Bilayer expts:
Clamping not entirely clear – what voltage was the bilayer clamped at? It is stated membrane voltage varied from – 10 mV to +20 mV. Was this in incremental steps? Was the polarity reversed at frequent intervals? The data reported in the figures all seems to be 0 mV?
Please clarify the sample sizes. In the figure legend for figure 1 it states ‘n = 8’. As far as I can tell, the only condition in which an n =8 is shown is 0 uM of Fig1E. For example, Fig 1B shows an n of 7 data points at 0 uM flecainide which decreases to 5 data points at 2 uM and thereafter. Consequently, I am wondering how statistical comparisons have been made – is the data paired as suggested by the repeated measures ANOVA described in the methodology? Similarly, in fig 1E the n is higher for 0 uM flecainide than the remaining doses – presumably this data is therefore not paired? Or some points have been omitted?
The same applies for Figure 2. Sample size is again listed as n = 8 yet in panel 2b most of the n seems to be 5, except for 50 uM which is n = 6, but n = 5 in 2C.
For the data in figure 4, n = 9 but this does not appear to be the case in the figure.
All the data has been presented in the normalised format – is there much variation in the absolute magnitude of open probability? I note that channels included vary from a starting point of 0.2 – 0.57 as stated in line 109. Is any variation in flecainide potency observed dependent on the relative level of channel activity? And how do you disentangle the block of countercurrent compared with any potential activation of RyR2? Salvage et al 2021 (PMID: 34440870) demonstrate an activating mechanism of flecainide. Is it possible this could be occurring here? If so would it confound the potential block reported? ‘In the absence of a tris gradient, the G versus flecainide relationship exhibited a very weak correlation, as evidenced by a Pearson correlation…..’ Could this be elaborated on please. Is flecainide exerting an effect on the SR lumen-to-cytosol current?
In line 131 absolute values of conductance are reported, can a table be provided of all the relevant values for easy reference please.
QCM experiments:
Please include some references to the background of this technique, particularly with respect to circuitry. Further information needed for the sensor as the company supplies a wide range of QCM sensors.
The section denoting preparation of liposomes should be mentioned before their addition to the experiment.
How are liposomes immobilised? Is it simply a matter of waiting an appropriate time for them to settle and adsorb to the surface? When is the 5 mM CaCl2 added? Could this be indicated on fig 5A please?
It is noted that flecainide was prepared as a 30 mM DMSO stock and then diluted as appropriate. I am wondering if you have controlled for the presence of DMSO in the experiments of figure 5? Could this be interacting with the membrane or affecting membrane integrity given that it readily crosses tissue membranes? (Jacob SW et al 1986, PMID: 3007027).
At what point after lipid formation was the 0 uM value recorded? And how long was each dose of flecainide/caffeine applied for. The highlighted lipid membrane formation segment is broad and variable – visually, according to the trace in A once the membrane has stabilised there appears to be little variation between this starting point and the addition of 0.2 uM.
Is it possible to incorporate RyR2 channels into the liposomes? If so, could anything be inferred about the direct interaction of flecainide? I am just wondering whether specific changes in frequency might be observed that differ from those seen when flecainide interacts with the lipid membrane – would it theoretically be possible to observe binding and unbinding of flecainide? This would be interesting given the mode of fast block.
There are some contradictory arguments. In lines 400 -403 it is state that ‘..Bannister et al challenged the view that the coexistence of opposing current and countercurrent could stabilise flecainide in the RyR2 conducting pore, and our results clearly validate this concept’ (do you mean your results validate Bannisters concept, or the concept that opposing current and countercurrent could stabilise flecainide..?)
Yet, at lines 413 – 416, it is stated ‘..we can speculate that while it is highly likely that opposing current and countercurrent flowing simultaneously through the RyR2 channel stabilise flecainide binding in the RyR2 pore, they concurrently may cause a significant increase in the frequency of short-lived blocking events.
Round 2
Reviewer 1 Report
Comments and Suggestions for Authors
The authors have performed additional experiments with DMSO and have satisfactorily answered my questions and concerns.
Reviewer 2 Report
Comments and Suggestions for Authors
The authors have satisfactorily answered most of my queries. However, there are still some major outstanding points.
Point 6. Regarding sample sizes and statistics.
This reviewer does not agree that data imputation is appropriate here and certainly is not validated by the desire to perform a repeated measures ANOVA. As it stands, the data is categorically not repeated measures over the range of concentrations tested.
I note the reference the authors have used to validate this approach. I would like to point out that noted in this is that RM ANOVA makes 3 key assumptions:
The continuous dependent variable is approximately normally distributed
The categorical independent variable has three or more levels
No outliers in any of the repeated measurements, and sphericity.
This has not been addressed by the authors. In any case, the authors are not limited by missing clinical samples etc. The experiments should have been designed in a manner that would allow repeated measures ANOVA, or an alternative approach to the statistical analysis needs to be undertaken.
QCM experiments
Point 14. Regarding the effect of DMSO, this was in relation to effects on these experiments not the bilayer experiments. Yet the authors carried out further experiments on channels rather than the quartz crystal microbalance. So, it is not apparent whether the effects in the QCM experiments can be entirely attributed to flecainide given that neither DMSO nor the time course of the experiment are accounted for.
Point 15. It would therefore seem that the representative trace and aggregated data in figure 5B do not match. -Δfs at 0.2 appears to be a value of approximately 5. But the dip after 0 uM in the trace occurs while still in 0 uM. Then following the trace while 0.2 uM flecainide shows minimal variation from the 0 uM value. It would seem the authors have incorporated the end of signal indicating lipid membrane formation – recorded data should be after this.
Round 3
Reviewer 2 Report
Comments and Suggestions for Authors
This is a nice addition to the field and the efforts of the authors in addressing my comments are appreciated.